# Use of Robotic Platforms as a Tool to Support STEM and Physical Education in Developed Countries: A Descriptive Analysis

**DOI:** 10.3390/s22031037

**Published:** 2022-01-28

**Authors:** Pedro Ponce, Christian Fernando López-Orozco, Germán E. Baltazar Reyes, Edgar Lopez-Caudana, Nancy Mazon Parra, Arturo Molina

**Affiliations:** 1Writing Lab, TecLabs Tecnologico de Monterrey, Mexico City 14380, Mexico; pedro.ponce@tec.mx; 2School of Engineering and Sciences, Institute for the Future of Education, Tecnologico de Monterrey, Mexico City 14380, Mexico; a01631685@itesm.mx; 3School of Engineering and Sciences, Tecnologico de Monterrey, Mexico City 14380, Mexico; a01331329@itesm.mx; 4Faculty of Psychology, Universidad Nacional Autonoma de Mexico, Mexico City 04510, Mexico; nancymaz01@yahoo.com.mx; 5Tecnologico de Monterrey, Mexico City 14380, Mexico; armolina@tec.mx

**Keywords:** social robotics, assistive education, robot NAO, LEGO^®^, elementary school, STEM, educational innovation, higher education

## Abstract

The lack of interest of children at school is one of the biggest problems that Mexican education faces. Two important factors causing this lack of interest are the predominant methodology used in Mexican schools and the technology as a barrier for attention. The methodology that institutions have followed has become an issue because of its very traditional approach, with the professor giving all the theoretical material to the students while they listen and memorize the contents, and, if we add the issue of the growing access to technological devices for students, children carrying a phone are more likely to be distracted. This study aims to integrate technology through assistive robots as a beneficial tool for educators, in order to improve the attention span of students by making the learning process in multiple areas of the Mexican curriculum more dynamic, therefore obtaining better results. To prove this, four different approaches were implemented; three in elementary schools and one in higher education: the LEGO^®^ robotic kit and the NAO robot for STEM (science, technology, engineering, and mathematics) teaching, the NAO robot for physical education (PE), and the PhantomX Hexapod, respectively. Each of these technological approaches was applied by considering both control and experimental groups, in order to compare the data and provide conclusions. Finally, this study proves that the attention span is indeed improved as a result of implementing robotic platforms during the teaching process, allowing the children to become more motivated during their PE class and become more proactive and retain more information during their STEM classes.

## 1. Introduction

Professionals in education are confronting a constantly increasing lack of attention and exhibition of behavioral problems in the classroom by their students. The traditional approach used in schools, where the professor gives all the theoretical explanations to the students while they merely listen and memorize the content given, makes students lose interest in their classes, affecting their academic performance [1]. The lack of more dynamic approaches negatively influences the children’s ability to comprehend theoretical concepts when it comes to indoor classes, and the usual outdoor physical education activities are inefficient and discouraging for the students, placing Mexico as the most inactive country in levels of physical sport activities [2].

The use of these traditional approaches in the different areas of the Mexican curricula affects the children’s motivation to develop physical activities, and also their design, reasoning, and measurement competencies, making it more challenging for them to understand future topics.

In order to improve the learning and cognitive processes (such as memory and executory functions), educators must understand that it is fundamental that the attention of children in the first years of school is stimulated [3], and to achieve that, there is a need to develop open-science strategies to guarantee the improvement of educational achievements among the school-age population. The Educational Secretary of Mexico (SEP) has started implementing, in elementary and high schools, the use of information and communications technologies (ICTs), with the aim of generating greater inclusion with the students during their classes and creating more compelling explanations and exercises that could help students retain the information.

Educational robotics is a discipline whose objective is the conception, creation, and functionality of robotic prototypes, utilizing specialized programs to achieve pedagogic ends. For this study, a robotic platform will be defined as the use of the LEGO^®^ robotic kit, the use of the NAO robot, and the use of the PhantomX Hexapod. To evaluate and measure the impact that these platforms have, we used observational scales.

## 2. Information and Communications Technologies (ICTs) in Education

The Mexican government divides educational levels into four levels created and supervised by the SEP [4] as follows: Starter education. Attended by students younger than 6 years.Basic education. An educational period that teaches basic concepts to children between 6 and 15 years old.High school. Preparatory education taught before entering the professional studies.Superior education. Education focused on a specific area for the student’s professional life and possible postgraduate studies.

The first ICT used with educational purposes was implemented by Seymour Papert with the LOGO programming language [5] to teach mathematics. Nonetheless, in the technologically advanced era we live in, the use of computers has become less efficient with younger generations due to their familiarity with such technologies, so there is a need to innovate and adapt the tools to generate higher impact and capture the attention of students.

ICTs are increasingly used in different social sectors worldwide. In Mexico, where ICTs are only used by under 30 percent of the population, the weight of technology is evergrowing, allowing the student to provide efficient responses to challenges in the everchanging environments of the contemporary world [6]. Thanks to the Integral Reform of Basic Education, SEP started to invest in the distribution and use of ICTs with the objective of promoting “learning by reception” methodologies. These methods guide active and participative construction of knowledge from the students [7].

As educators, it is highly important to understand that an educational robotics proposal must be implemented after considering the learning environment, the planning of activities, the resources, the necessary time for the conclusion of each exercise, and the methodology [8] in order to obtain positive results.

This paper shows the process and result of the implementation of the LEGO EV3 platform for STEM-oriented tasks in the basic education level and the implementation of the NAO robot for physical education, with students from the 4th, 5th, and 6th grades of elementary schools in Mexico. In addition, two different studies for STEM education were implemented using both the Hexapod robot, with undergraduate students, and the NAO robot, with elementary school students, whose aim was to measure the perception of students towards the robots.

Even though the three different approaches mentioned before are completely independent from one another, this study has the aim to study the influence of robotic platforms with different curricula and educational levels.

## 3. Assistive Robots in Education

Robots have evolved into a facilitatory tool with significant achievements in the learning process, being incorporated into a relevant strategy for motivating children and increasing their curiosity [9].

There are plenty of studies describing the importance of social robotics [10], but there are still no models showing the robot as a tool to obtain better results in the teaching–learning process by itself. Some successful studies involve disruptive intervention of a robot that helps to lower anxiety [11], robots as additional technological tools to generate an adequate pedagogical scenario [12], and even evidence on how students feel more comfortable and solve exercises more quickly when a robotic platform is implemented during the class sessions [13].

Some of the studies previously made to prove the benefits of using a robotic platform in STEM and physical education (PE) are shown in [14,15], where, in the first study, they used the LEGO Mindstorms robotics kits to introduce children to robotics and technology applications, and in the second one, they used the socially-assistive robot (SAR) to improve the motivation of elderly people for physical exercises. The NAO robot was also used for a different study to give dance lessons to a group of children, explaining all the movements that they needed to learn and perform a dancing routine [16].

For the case of undergraduate students and robotics to support education, most of the studies focus on technical implementations and artificial intelligence. A study [17] showed the implementation of four robots connected (three son robots and one mom robot) to perform specific tasks, allowing students to learn about the use of sensors and communications. In addition, Ref. [18] shows a different approach to support education, where the professor gives access to multiple robotic platforms for the students to program and increases the interest in the topics. In contrast to these studies, the proposal of this project focuses on the robot as a support for the professor, explaining some topics and leading the dynamic of interaction. This approach has a both-sided interaction with the students, giving a different approach to learning while directly sharing the topics instead of the student overseeing the programming of the platform.

## 4. Attention Span and Its Measurement in the Classroom

For this study, attention refers to how we actively process a fraction of a vast number of stimuli using our senses and other cognitive processes, and five indicators are considered to help to measure the concentration of a person: precision and memorization of information, the habituation to a stimulus, the dishabituation of such stimulus, the distraction or the neglect of the main activity, and the motivation and enthusiasm shown when working in a specific task [19].

The first step to measure the attention span of children is to begin with an observational methodology, create unique observational instruments [20], and have the scientific knowledge to study the occurrence of perceptible behaviors in a way that can be adequately registered and quantified [21]. Measuring attention can be really challenging without a concrete structure, so it is highly important to create an assertive methodology and follow it step by step.

## 5. Description of the Robotic Platforms

It is important to describe the characteristics of the LEGO robotic kit, the PhantomX Hexapod, and the NAO robot, the three robotic platforms used for this study. The LEGO robotic kit and the NAO robot were adapted considering the contents established from the Educational Secretary of Mexico (SEP) for students of 4th, 5^th^, and 6th grades of elementary school (basic education), while the PhantomX Hexapod was adapted considering contents from their STEM class at the university.

### 5.1. LEGO^®^ EV3

The version of the LEGO^®^ Mindstorms kit EV3 used allows the user to assemble the typical LEGO^®^ pieces with multiple sensors and actuators to create robotic interfaces that are capable of interacting with its surroundings.

With a total of 541 pieces, the educational kit makes it possible for the students to design, build, and program hands-on robotic projects, linking them with mathematical and scientific concepts.

The ability of the EV3 kit to be programmed through NI LabVIEW expands the usability of the platform, making it possible to develop even more complex designs that allow professors to apply specific concepts, using the robotic interface built by the student.

### 5.2. NAO Robot

The NAO robot has 25 degrees of freedom (DOF), which gives it the capacity to execute a wide range of movements such as walking, sitting, standing, dancing, evading obstacles, kicking, and grabbing, among others.

Integrated with WIFI, the NAO is entirely autonomous and can establish a secure connection to the Internet to download and transmit content. Equipped with two speakers and microphones, it has a quality system that can reproduce music; its voice recognition localizes the origin of sounds so that it can turn its head towards the source.

This humanoid robot was programmed in the visual environment Choregraphe as well as in Python, depending on the programmed routine [22,23].

### 5.3. PhantomX Hexapod

The PhantomX AX MKII Hexapod robot is an open-source platform in the industry, meaning that all 3D CAD, electronics, and the programming software are open for everyone. 

It is a six-legged robot with 3 degrees of freedom in each leg, it is compatible with Arduino, and it has been an amazing tool not just for hobby, but for education and research as well.

## 6. Class Preparation

A different school interacted with each robotic platform, depending on the case of study. Three different elementary schools took use of two of the platforms, the *LEGO^®^ EV3* for STEM classes and NAO robot for PE and STEM classes. Additionally, a hexapod robot Phantom X AX Mark II was used for an undergraduate course, allowing the study to obtain different perspectives in terms of age and educational levels.

### 6.1. STEM Classes with the LEGO^®^ EV3

A total of 24 mathematics and science tasks, shown in Table 1, were designed and implemented in the elementary school, each one following four steps for its correct implementation:

First, the theoretical concepts were given inside the classroom by the professor to introduce the students to the concepts required to understand the material. A total of 54 students were interacting the same way at this point of the study, with the introductory and theoretical concepts, but for the second phase, just 12 students were randomly selected to have a small sector of the group interact with the robot. This split was made to obtain a comparison before, during, and after the interaction, and in that way we could measure the technological approach in comparison to the 42 students that continued with the traditional way of learning.

The second part consisted of building the robotic platform that would help every student selected give a practical approach to the topic. For this part of the task, every student had to form work groups to organize the building of the sensors, actuators, and the interface of the EV3 (referred to as “brick”). Every work group had a manual with the sensors and actuators required, as well as a clear explanation of each port and how they needed to connect them. Every group had the liberty to use any pieces they wanted to give form to the platform, as long as the connections given in the manual were followed.

The third part consisted of running the preloaded program of the brick. By following a series of instructions, every student was able to interact with the robotic platform while learning the theoretical concept seen in class. For the students to accurately relate the exercises to the class concepts, the tasks challenged them to apply the key concepts to answer multiple questions that the program would give them during the interaction.

Finally, the last part of the task consisted of evaluating the concepts reviewed during the interaction with the EV3 kit. The evaluation was made through a questionnaire designed using NI LabVIEW. In these questions, every student was asked to individually solve multiple problems related to the concepts seen during both the task and the class. After solving all the questions, the system would use a fuzzification process to tell the student which parts required further study to improve his results; this process, shown in Figure 1, makes it possible to determine in which exercise or example every student requires further study in order to improve their results. A detailed explanation of each task is given in Table 2 where, even though it only shows 12 topics, each one could have been adapted differently for different school years.

This program also creates a single document for the professor showing the numerical results of every student, to make it easier to keep the surveillance of the progress of the whole class. This implementation and a further explanation of the model of evaluation is described in [24].

### 6.2. Use of a NAO Robot for PE Class

The recommendations made by a specialist in physical education (PE) were taken in order to prepare the activities, evaluating the adequacy of the environment [25]. At the end of the planning, it was decided that every session would consist of 10 min of warm-up, following the routine shown in Table 3, followed by the exercises shown in Table 4, appropriate to the age groups and following from three to five repetitions of the exercise sequence. Some integration and socializing exercises were included at the end of the session to support the observations of the *SEP*. In Figure 2, a block diagram of the entire process is shown, from the class planning to the analysis of results.

The warm-up needed to be progressive, with the intensity of the exercises changing gradually, as observed in Figure 3. The duration of the warm-up activities needed to last no more than 10 min per hour of class. It was intended to exercise all the muscle groups before continuing with the rest of the activities. Finally, the specific movements and activities were detailed.

The second and most crucial phase increases the difficulty of the exercises drastically, and the children pay more attention to the activities.

The proposed exercises were selected from the SEP’s recommendations after an interview [22] and a consultation with a physical education specialist. Those exercises, shown in Table 4, were authorized by the elementary school teachers, and were implemented depending on the group’s motivation or show of resistance to perform them. If the majority of the group would be motivated, the robot would ask them to repeat the sequence.

The specifications of the *SEP* govern the auxiliary model for the physical activities for schools. The routine is made up of two phases: the first one is the warm-up, which is of great importance for the development of the class and the children’s health; the second one consists of a more complex exercise routine. The complete explanation of each routine and exercise is described in [23,27].

### 6.3. Use of a NAO Robot for a STEM Class

For this case of study, the NAO robot interacted with students between 3rd and 6th grade of elementary school. For every grade, the groups of 44, 46, 48, and 48 students, respectively, were divided into two groups of the same number each: one that interacted with the robot (see Figure 4), and another one that did not. The idea of this approach was to compare the differences in the attention span of the students when performing the same exercises and activities with and without the platform. During each session, a different STEM topic was given to the students, covering three different activities that discussed sound propagation, the metric system, and fractions with whole numbers.

For the group that had no interaction with the robot, the professor was in charge of giving the explanations and activities to the students, while the robot was the one in charge of talking with the students in the other group. To prevent additional noise in the observations, both groups worked with a given script with the corresponding explanations and activity rules. In this way, the attention span evaluation could be focused on the way the students respond to the same information, given by different members (professor or robot). Additional to the attention span, the academic performance of each group was also evaluated, with an evaluation before and after the interaction with the robot.

In these sessions, the attention span of the students was observed and evaluated through an observation protocol applied by a group of psychology students based on the methodology proposed by Sternberg [19]. The protocol evaluated a total of seven sections of the student’s attention span, described in Table 5. At the same time, a usability survey was conducted among the school professors and the technical team who operated the robot, shown in Table 6. The usability test is better described in [26].

### 6.4. Use of the PhantomX Hexapod for a STEM Class

The implementation of this platform consisted of combining the use of a Hexapod Phantom X AX Mark II robot with an Arduino card, sensors, and a user manual with different exercises for the student, as shown in Figure 5. This platform was given to a group of first-semester undergraduate students of the department of mechatronics. Each class and session were progressively advanced from familiarization with the platform and its sensors to using them for learning different path-planning algorithms (from search algorithms to roadmaps and cell decomposition).

At the end of the course, a user-experience survey was given to the students to evaluate the contents and activities performed with the robotic platform, as well as the performance of the professor during each class. A better description of each topic covered with this platform is shown in [26].

## 7. Analysis of Results

The analysis made for each study was based on a questionnaire that the observers completed at every session and was compared with the results of each scholar period in term for the indoor classes.

Table 7 shows the evaluations from the STEM classes of every child that participated in the evaluation. The ones marked in yellow are the students that interacted with the LEGO EV3 kit before, during, and after the robotic platform was implemented, while the others are from the students that only interacted with their professor in the traditional way.

The results obtained from this application show that every student that interacted with the robotic platform at least maintained a positive grade, and even improved their general grades compared to the students that followed only the traditional class model with the professor. At the same time, the professors noticed an increase in the student’s proactivity when using the robotic platform. Separating the mean grades of the students that interacted with the robots from the ones that did, it can be observed that before the implementation of the platform, the difference in grades was not so representative (8.2857 for students that did not interact and 8.6667 for the ones that did). However, during the implementation of the robotic platform, the students of the traditional model obtained a mean grade of 7.9643, while the ones that worked with the EV3 kit obtained a score of 8.5. Once the implementation of the kit ended, the traditional model achieved a mean grade of 8.4048, while the students that worked with the kit had a mean score of 9.0417. These results prove that the use of a robotic platform improves the retention of information, as well as the academic performance of the students.

For the evaluation of the PE class with the NAO robot, a qualitative study of comments made by the observers (mostly psychology students) was implemented. This observation protocol showed relevant observations with regards to the points of interest (motivation and attention) during the PE sessions. A brief description of this study is shown in Table 7, where the yellow rows represent students that worked with the robot, and it was required in order to gather data and to be able to show results beyond a numerical scale, considering the behavior of the children during the entire class. The qualitative analyses were made by considering both scenarios, with and without the robot. We invite reading [27] to observe the complete analysis.

In the case of evaluating the usability of the NAO robot for the STEM classes in elementary school, the final scores of each group were compared to analyze the performance of each of the students, as well as the usability of the platform. As demonstrated in Figure 6, the general performance of the students that interacted with the robot showed that the use of a robotic platform positively influences the attention span of the student, improving their general performance during class, as shown in the evaluations results. Figure 7 and Figure 8, similarly to the results shown in Table 8, show how the students prefer an educational environment with a robotic platform that allows reinforcement of the theoretical concepts seen in class. The acceptance of the class model while using a robotic platform is demonstrated to be more efficient and desired by the students, improving their academic life.

Both figures show the results obtained after the evaluation mentioned in Table 5, where the psychology students gave the corresponding scores to the groups and made an average of the different interactions to obtain a single metric to present. Overall, the results of both evaluations show a predominance in the group that interacted with the NAO; both positive and negative features show a patterns towards acceptance of the NAO robot, but a slight difference can also be observed in the metric system and fraction results.

Regarding the evaluation of the undergraduate course, the general results are shown in Table 9, where it is also demonstrated that the use of a robotic platform improves the quality of the class, making it more likeable for the students, while assuring a good performance during the class. For further results and analysis, Ponce et al. describes the complete structure of the results [26].

## 8. Discussion

Even though the four methods presented in this work were completely independent from one another, this discussion will be centered in the improvement shown in the benefits of using robotics to support education.

Two main measurements were considered in the study: a measurement for the attention span or interest of the students, with the studies of the NAO robot for PE and the PhantomX Hexapod for an undergraduate class, and a measurement for the academic improvements or grades of the students, using the NAO and the LEGO EV3 platforms both for STEM classes.

The first measurement was analyzed considering a qualitative analysis used for the PE class with the NAO robot and a survey applied directly to the undergraduate students after using the PhantomX Hexapod. Both studies, shown in Table 8 and Table 9 respectively, show that there is indeed an increase of the attention span of the students or the likeability of the robot when it comes to supporting their education. As for the PE class, the students clearly show more interest and excitement while interacting with the robot, while for the case of undergraduate students, they showed a high percentage of satisfaction through the survey.

The second measurement, the academic performance of the classes, was measured directly from the grades obtained before, during, and after the interaction with the LEGO EV3 used for a STEM class, while for the case of the NAO robot, used for a STEM class as well, was measured by applying a test before and after their interaction. Both results, shown in Table 7 and Figure 6, respectively, show a slight difference between the results obtained with and without the robot. Those two studies could not be enough to make higher conclusions in terms of the material learned through the robots, but it does help to mention that for both cases the students showed more proactivity and interest during the entire session, shown in Figure 7 and Figure 8, which helps to support the previous measurement analyzed.

The four methodologies studied allowed this work to focus on the importance of the robotic platforms to improve the attention span of the students. A novel approach for the students, such as a robotic platform, can lead to more dynamic classes with higher rates of interaction between professors and students. There is still room for improvement in terms of the method of teaching for the students to retain the information, but a first step to capture student’s interest could be achieved with the review of this work. 

## 9. Conclusions

The convenience of using robotic platforms to help the learning process, increase attention levels, and motivate elementary-school-aged children was shown. This study demonstrated an increase in the attention span, meaning that the robotic system is an appropriate step in a global strategy to widen the perspective on the treatment of this problem in Mexican children.

Although the quantitative results proved there was not a big difference between the students that used the ICT and the ones who do not, it is essential to mark the fact that the use of the EV3 kit allowed the students to be more proactive during their classes. This fact made them participate more and answer more vividly the questions given to them during class.

From the results shown in this project, it was observed that the robotic platform increased the attention span, especially of children with low academic performance, when it comes to physical education classes, and it also gives the student an opportunity to learn at a different rhythm, making it easier for them to find a direct implementation of the abstract concepts seen with no practical implementation during a regular indoor class.

The best results were obtained in the dimension of interest in the task, with motivation, as much as enthusiasm, notably observing the highest levels during the sessions in both the analysis of the group and that of the focal children.

This study also allowed us to establish that children with symptoms that could indicate small attention spans showed a significant improvement in their motivation and attention, which was recorded in the two types of analyses that were carried out using psychological tools designed for this assessment.

## Figures and Tables

**Figure 1 sensors-22-01037-f001:**
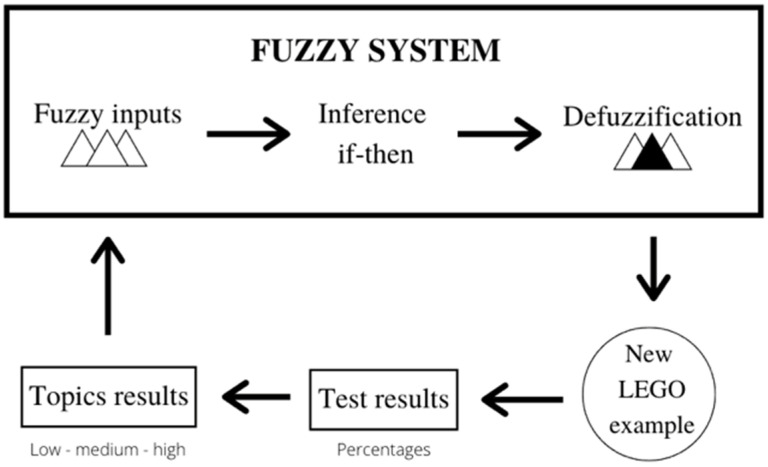
Fuzzy topology generated for teaching STEM task.

**Figure 2 sensors-22-01037-f002:**
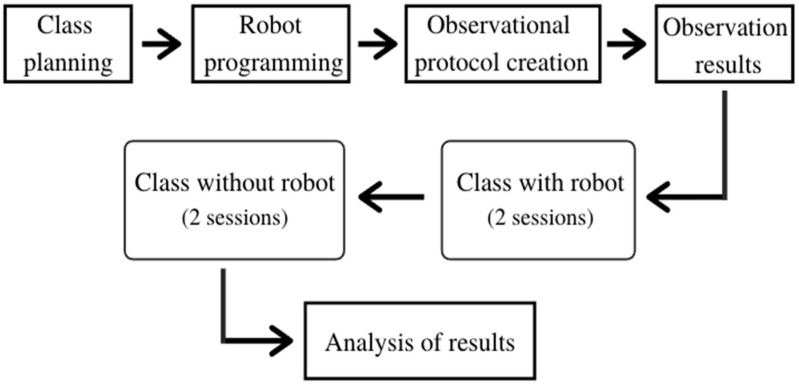
Project block diagram.

**Figure 3 sensors-22-01037-f003:**
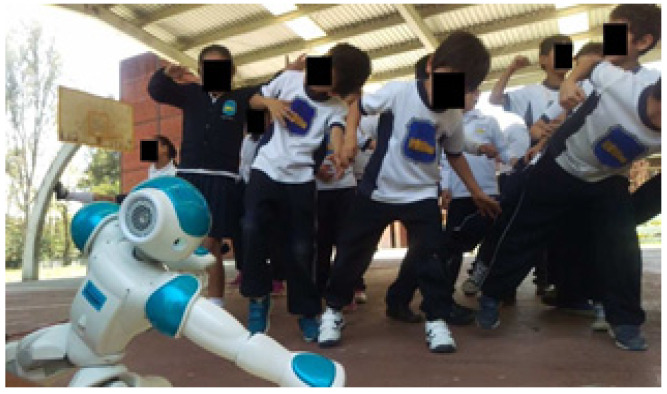
Use of the NAO robot in a PE class [26].

**Figure 4 sensors-22-01037-f004:**
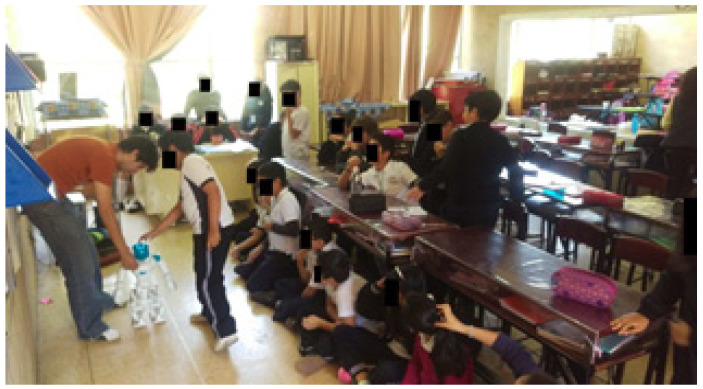
Use of the NAO robot in a STEM class [27].

**Figure 5 sensors-22-01037-f005:**
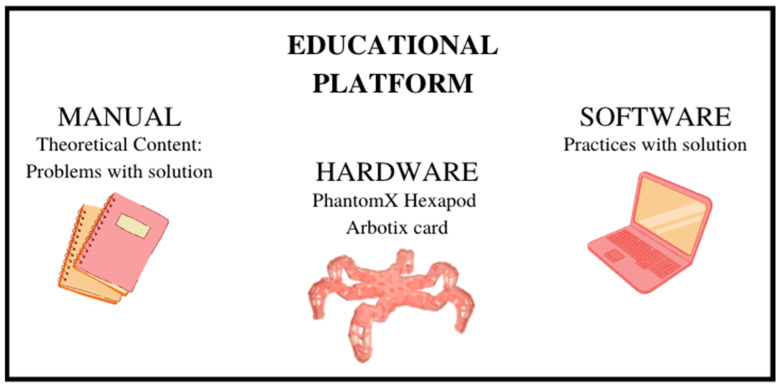
Hexapod platform for the undergraduate course.

**Figure 6 sensors-22-01037-f006:**
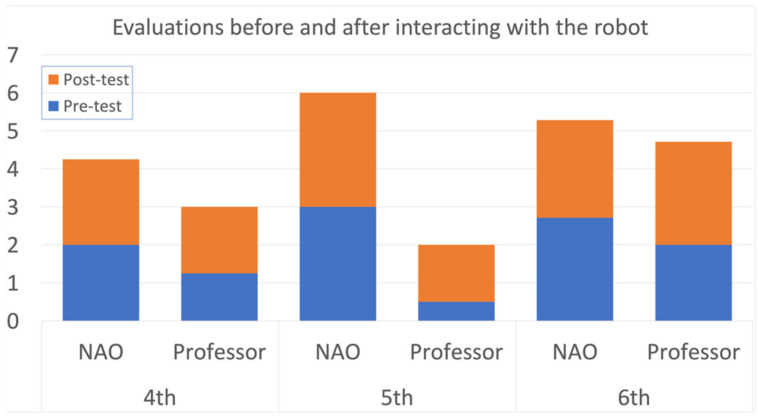
Comparison of results in the STEM class with and without the NAO robot in 4th, 5th, and 6th grade.

**Figure 7 sensors-22-01037-f007:**
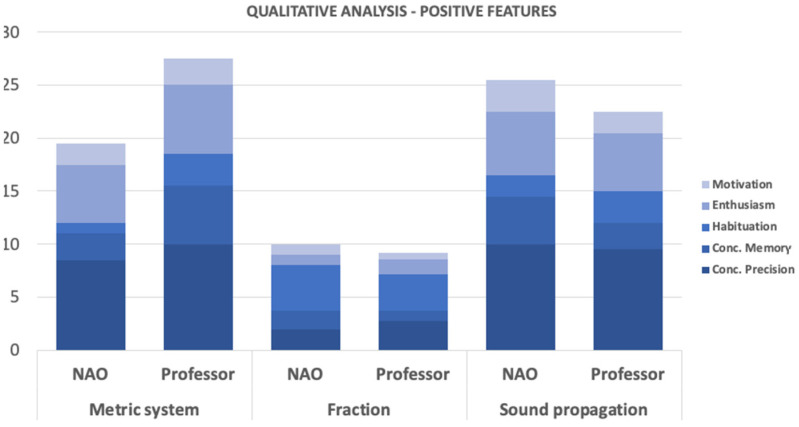
Qualitative analysis in the STEM class with and without the NAO robot—positive features.

**Figure 8 sensors-22-01037-f008:**
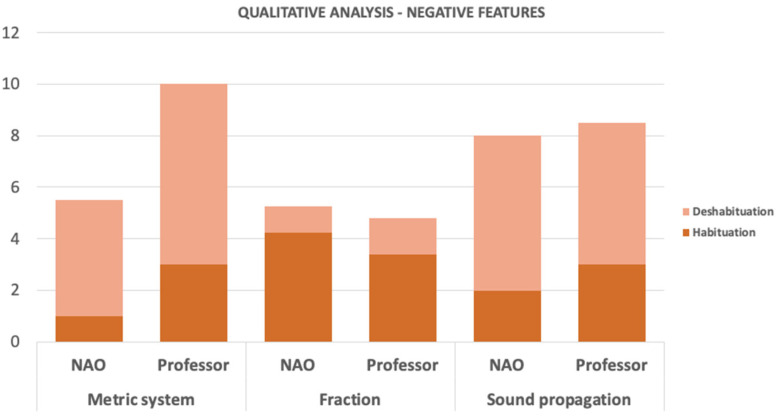
Qualitative analysis in the STEM class with and without the NAO robot—negative features.

**Table 1 sensors-22-01037-t001:** Tasks distribution.

4th Grade	5th Grade	6th Grade
Angles	Angles	Angles
Equivalent fractions	Numeric line	Cartesian map
Area (Rectangle)	Area (parallelograms)	Area (circles)
Perimeter	Perimeter	Perimeter
Identify figures	Identify figures	Percentages
Fraction operations	Proportions	Divisibility
Metric System Conversion	Parallel and secant lines	Negative numbers
Conversion from fractions to decimals	Conversions of the Metric System	Conversions of the Metric System

**Table 2 sensors-22-01037-t002:** Description of the LEGO^®^ EV3 tasks.

Topic	Description
Angles	This task was made by building the arms of a clock-like interface using the EV3 kits servo motor, with a cardboard clock previously designed where different animals were located instead of the numbers of a regular clock. The difficulty of the tasks increased according to the grade of the student.
Equivalent fractions	Consisted of using the EV3 color sensor to save a certain number of objects of different colors. Each color represented a specific fraction, which was added to a total sum every time the student recorded it into the brick. After recording each color individually, it asked the student to calculate the sum of two different colors.
Conversion from fractions to decimals	For this task, the students would record pieces of different colors, where each one would represent a certain amount of weight. After recording many objects from different colors, the brick asked the students to represent, using fractions, the total weight registered of each color.
Geometric figures	The students needed to build a kart using the EV3 and placing it with a marker on a piece of paper. Depending on the grade, the students needed to draw different types of parallelograms, triangles, and circles. The task was to identify which type of figure was in the specific case, and measure and obtain both the perimeter and area of the figure, comparing their results with the answer shown by the brick.
Metric System Conversion	After building a kart with the EV3 kit, every team needed to move it to a specific distance shown in the display of the brick. Once the kart moved, they needed to compare such distance with a meter and make the comparisons.
Number line	The previously built EV3 kart would move to a certain distance and would ask the students to obtain the distance by giving them two options. The intention is that by using only a meter as a reference, the students should be able to convert to the units displayed in the brick to select the correct answer.
Parallel and secant lines	This task consisted of drawing multiple lines, both parallel and secant, using two different colors. The EV3 kart would follow one line, and if it detected another color, then it would indicate the presence of a secant line.
Proportions	This task required the student to record a certain number of objects of different colors. Every color recorded was assigned to be equal to a certain number of pieces of another color. After recording every piece of every color, the brick asked the students how many pieces of a specific color were required to obtain the amount of another color.
Negative numbers	The EV3 kart would move right (simulating positive values) and left (simulating negative values). During the interaction, the brick asked the students in which position it would end up when adding a certain positive number with a negative one. For this case, the students had a visual way of comprehending how negative numbers work, as well as their existence and use in multiple mathematical applications.
Cartesian map	This task would move in four directions according to the instruction given. The intention was for the students to determine how many times the robot should move upwards, downwards, left, or right to reach an object located in another position of the map. The professor was also able to install some obstacles in any place of the map in order to encourage the students to create more complex routes for reaching the goal.
Divisibility	This task works as a contest between different teams of the class. Once each team built an EV3 kart, they were asked to place them at the beginning of a board. In turns, every team answered which number between three options was divisible by the number shown in the brick. If the answer was correct, the kart moved to the next stage; if not, the kart stayed at the same place, waiting for their next round.
Percentages	This task required the student to record a certain number of objects of different colors. After recording all the colors, the brick asks the student which color is the one that covers a specific percentage of the total colors registered. In this way, the students need to make the calculations to find the answer.

**Table 3 sensors-22-01037-t003:** Warm-up exercises and time.

Warm-Up	Time
Neck	1 min
Shoulders	1 min
Arms	1 min
Waist	1 min
Legs	30 s
Feet	30 s
Light Jog	5 min

**Table 4 sensors-22-01037-t004:** Exercises sequence.

Exercise	Time
Squats	20 s
Crunches	15 s
Jumps	20 s
Burpees	10 s
Lunges	10 s
Push-ups	15 s
Mountain climbers	20 s
Side bends	15 s
Relays	3 s

**Table 5 sensors-22-01037-t005:** Question distribution for the observational scale.

Dimension	Question Numbers	Number of Questions
Concentration: Precision	1–9	9
Concentration: Memory	10–12	3
Habituation	13–22	10
Dishabituation	23–26	4
Distraction	27–32	6
Interest in the task: Enthusiasm	33–36	4
Interest in the task: Motivation	37–41	5

**Table 6 sensors-22-01037-t006:** Usability questionnaire.

	Questions
1	The robot needs to keep the user informed about what is happening through reasonable feedback.
2	The robot speaks the user’s language with familiar words, phrases, and concepts instead of in system-oriented terms. It follows real-world conventions, giving the information in a natural and logical way.
3	The users must be free to select and sequence the activities, instead of doing what the robot tells them to do.
4	The users must not ask if different words, situations, or actions have the same meaning. They must follow the robotic platform conventions.
5	The error messages must be expressed in a simple language (without codes).
6	The robot does not present programming errors that lead to a malfunction during the activity.
7	The user must not remember all the information of the task. The materials and robot’s actions and options must be visible or clearly visible when needed.
8	The robot’s dialogs do not have irrelevant or strange information.
9	Even though it is better to use the robotic platform with documentation, the use is intuitive.
10	The robot can endure, extend, complement, or improve the user’s abilities, basic knowledge, and experience. This motivates its use.
11	The human–robot interaction improves the quality of your labor life.
12	The robot helps the user to protect its personal or private information.

**Table 7 sensors-22-01037-t007:** Fourth grade students’ scores.

ID	Bimester before Using the EV3 Kit	Bimester Using the EV3 Kit	Bimester after Using the EV3 Kit	Student’s Final Score
1	8	8	9	8.80
2	10	9	8	8.83
3	10	4	8	6.96
4	4	9.5	9	7.82
5	10	9	9	9.14
6	9	7	9	8.80
7	9	6	7	7.40
8	9	6	8	8.46
9	9	9	10	9.15
10	8	7	8	7.66
11	10	10	9	9.00
12	9	10	7.5	8.96
13	8	6	10	8.66
14	7	7	10	8.52
15	9	6	6	6.86
16	8	8	7	7.22
17	8	7	9	8.60
18	10	9	10	9.52
19	10	10	8	9.00
20	9	10	10	9.46
21	8	8	10	9.06
22	9	9	10	8.72
23	9	8	8	8.46
24	8	10	8	8.72
25	10	10	10	9.66
26	10	10	10	9.92
27	9	10	9	9.40
28	10	9	10	9.80
29	9	9	10	8.40
30	9	7	10	8.60
31	8	10	10	9.26
32	7	10	8	8.26
33	9	9	10	9.20
34	10	10	10	9.52
35	9	9	9	9.26
36	7	9	9	8.46
37	8	8	9	7.76
38	7	7	5.5	7.02
39	8	8	6	6.96
40	9	9	9	8.42
41	8	8	8	8.26
42	9	8	10	8.86
43	9	8	9	8.46
44	7	7	6	6.42
45	7	8	8	7.86
46	4	4	6	5.40
47	7	9	8.5	8.56
48	9	8	10	9.25
49	8	8	8.5	8.36
50	8	3	5	6.12
51	7	4	4.5	4.96
52	6	7	7.5	7.56
53	9	9	8.5	8.80
54	7	9	10	8.00

**Table 8 sensors-22-01037-t008:** Qualitative analysis for the PE class with the NAO robot.

Concentration and Precision	Group without the robot	Some students lacked enthusiasm when performing the exercises. The idea of competition motivated them.
	Group with the robot	In general, the group was always active, but the students became easily tired.
Concentration and Memory	Group without the robot	-
	Group with the robot	The exercises were performed more carefully.
Habituation	Group without the robot	The students yawned at the beginning of the class, but this became less frequent during the class.
	Group with the robot	The students complained that the exercise was too difficult for them.
Dishabituation	Group without the robot	The students never asked questions.
	Group with the robot	Some students asked questions insistently.
Distraction	Group without the robot	Most of the students lagged while performing the exercises.
	Group with the robot	In previous sessions, the students fell behind with the exercises. However, their attention improved with the robot and tried to perform the routines with the NAO robot.
Interest and Enthusiasm	Group without the robot	The students showed fun during the relay race.
	Group with the robot	The students imitated the robot to master the exercises.

**Table 9 sensors-22-01037-t009:** Survey results of the undergraduate course.

Generally, how do you grade the course?				
Very Satisfactory	Satisfactory	Indifferent	Bad	Very bad
20%	50%	30%	0%	0%
How do you evaluate the content of the course?				
Excellent	Good	Regular	Bad	Very bad
15.79%	52.63%	21.05%	10.53%	0%
Generally, how do you grade the performance of the instructor?				
Excellent	Good	Regular	Bad	Very bad
17.60%	50%	12.60%	0%	0%

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
