# Peer review of "Use of Robotic Platforms as a Tool to Support STEM and Physical Education in Developed Countries: A Descriptive Analysis"

_sensors, 2022, doi:10.3390/s22031037_

Round 1

Reviewer 1 Report

This study supports STEM and Physical Education in elementary schools by using robotics platforms, particularly LEGO robotic kit for STEM and NAO robot for  PE. The authors concluded that the robotic platforms enhanced children's attention and are useful additional tools to assist the school's teaching process.
Stimulating educational processes and studying methods to improve school success and physical literacy education have been pivotal in recent years, and applying new technologies is nowadays a relevant and dutiful matter. I have been fascinated by this educational approach (unknown to me), and I would congratulate the authors for exploring some new strategies.
Nevertheless, the style of this study is quite different from my previous experiences as an author or reviewer. Therefore, suggestions will be provided in this sense, hoping the following comments might be helpful to improve the manuscript.
In my opinion, the paper is well written, and the introduction and the description of the robotic platforms are clear and complete. On the other hand, the results are somewhat speculative, as methods, analysis, and Discussion miss information to fully understand the procedures and results analysis. 
Some concerns arise about the design of the procedures: if the study aimed to investigate the effects of robotic supports in learning, why has a pre/post design or comparison between groups by applying a statistic approach not been performed? I have no reason to doubt that robotic supports enhance childrens' motivation, attention and cognitive functions, strengthening the teaching process. What is missing is a straightforward objective approach to support with more robust results the conclusions that are only speculative since missing statistical analysis.
In this vision, I was expected to find details about subjects, testing methods, and statistical analysis in the manuscript.
Does the data collected can be re-organized in this way? 
A statistical comparison should appear in the results (Fig. 6, in particular).
Otherwise, the authors should "soften" their conclusions, as not originating from a robust analysis and comparison.
In addition, please consider adding more information about the analysis of the tool to evaluate the effect of the platforms (the questionnaire). There are several recalls to references (e.g. line 219, 259, 275), but more information about the procedure should be described in the manuscript.

Introduction
line 87:  8 and 9 are the same reference
line 104: [16] is a study with elderly

Class preparation
I suggest revising the subjects' section. How many children per class? And per Group? The articulation is quite challenging to be followed, and synthesis of groups and comparisons is required to help the reader. It should correspond and be consistent with the outcomes of the results and with the discussion/conclusion.

Methods
please add information about the procedure of PE training: Fig. 4 refers to a sequence; how many times is it repeated? Are there breaks? How long? Do the execution of the exercises has been managed?

Observation protocol to grade students: please provide more information to better understand how the students were graded, as no validated tools to measure the platform' effect on enjoyment, attention or motivation were used.

Table 7
Not clear.  How to read it? Improvements in final score? The trend? This has not been discussed. 
What do the yellow rows stand for? Please add a detailed legend to the table.

Figure 6
Units are missing; statistics are missing. What do Groups 1 to 3 refer to? There are no Group demographics in the Methods section (participants).
The grouping by grade should correspond to a clear description in Methods.

The Discussion is missing: it should address the literature results about teaching strategies and their relationships to technology, motivation, physical activity.

I hope my comments help the Authors in the revision process.

Author Response

Thank you for your time in the revision of our work. We took the recommendations and changed or added what it was requested, and also we added some answers to the questions you made.
Hope this helps to have a better understanding of the work we are attempting to publish.

A statistical comparison should appear in the results (Fig. 6, in particular).
Otherwise, the authors should "soften" their conclusions, as not originating from a robust analysis and comparison.
The statistical comparison was added as suggested. The changes are going to be mentioned in the next requests with each individual suggestion.
In addition, please consider adding more information about the analysis of the tool to evaluate the effect of the platforms (the questionnaire). There are several recalls to references (e.g. line 219, 259, 275), but more information about the procedure should be described in the manuscript.

Introduction
line 87:  8 and 9 are the same reference
They were indeed the same, one of them was removed.

line 104: [16] is a study with elderly
It was changed to reference [15] because of the previous change, and it was a typo that is now corrected. It is a study made with elderly.

Class preparation
I suggest revising the subjects' section. How many children per class? And per Group? The articulation is quite challenging to be followed, and synthesis of groups and comparisons is required to help the reader. It should correspond and be consistent with the outcomes of the results and with the discussion/conclusion.
During the class preparation section the number of students were added and also the division into groups. More explanation from the results was also made considering the statistical analysis that you suggested. Hopefully it will help to have a better understanding of the work.

Methods
please add information about the procedure of PE training: Fig. 4 refers to a sequence; how many times is it repeated? Are there breaks? How long? Do the execution of the exercises has been managed?
It was added in the paper. The robot asked to repeat the sequence three times, and after that it would depend on the groups motivation to keep carrying the sequence.

Observation protocol to grade students: please provide more information to better understand how the students were graded, as no validated tools to measure the platform' effect on enjoyment, attention or motivation were used.
We used the scale mentioned before. More information was also added in the paper.

Table 7
Not clear.  How to read it? Improvements in final score? The trend? This has not been discussed. 
What do the yellow rows stand for? Please add a detailed legend to the table.
All was added, the explanation of each column and that the yellow rows mean that the student did have an interaction with the robot.

Figure 6
Units are missing; statistics are missing. What do Groups 1 to 3 refer to? There are no Group demographics in the Methods section (participants).
The grouping by grade should correspond to a clear description in Methods.
A deeper explanation was added. It is the usability scale, meaning the answers of students for wanting the robot to keep interacting during their classes.

The Discussion is missing: it should address the literature results about teaching strategies and their relationships to technology, motivation, physical activity.
Extra discussion was added both, in the introduction and in the preparation or explanation of the methods.

Thank you for your time and we will be awaiting any further comments you may have.

Reviewer 2 Report

Was an IRB completed? It is unusual to name the schools where data was collected, as this exposes research participants to being identified. 

The word "diligence" seems out of place. Was that the correct word choice?

What is the protocol designed by psychology students? Does it have reliability or validity data to back up its use? Why not use an existing, verified instrument? 

It seems like a lot of things were done and reported on here but not in a systematic way that adds to the literature. The focus of the research isn't super clear (research questions would help with this) and the connections of the results back to the literature are also lacking. What is your theoretical framework?

Author Response

Thank you for your time in the revision of our work. We took the recommendations and changed or added what it was requested, and also we added some answers to the questions you made.
Hope this helps to have a better understanding of the work we are attempting to publish.

Was an IRB completed? It is unusual to name the schools where data was collected, as this exposes research participants to being identified. 
We did have the permissions from school, but we still decided to remove the names of school to keep it confidential. 

The word "diligence" seems out of place. Was that the correct word choice?
All changes were made, the word "diligence" was replaced by "task"

What is the protocol designed by psychology students? Does it have reliability or validity data to back up its use? Why not use an existing, verified instrument? 
Yes, the study was backed up by the methodology mentioned in reference 17. Also, some extra explanation was added in the paper (line 256).

It seems like a lot of things were done and reported on here but not in a systematic way that adds to the literature. The focus of the research isn't super clear (research questions would help with this) and the connections of the results back to the literature are also lacking. What is your theoretical framework?
Literature reviews were added to support the study from line 116.
Also additional explanations were mention in the introduction section, regarding the aim of the project which is to measure the different platforms as individuals and create the conclusions from that. There is no chance of comparing or relating 100% one another, because they were implemented in different time periods and with different methodologies. But we do can create some evaluations in terms of the benefits or improvements when it comes to the classes.

Thank you for your time and we will be awaiting any further comments you may have.

Round 2

Reviewer 1 Report

I would thank the authors for their efforts in answering the comments of the previous revision.
They addressed most of the issues and revised the manuscript.
I still have some concerns regarding the soundness of the data analysis on which the conclusions are based. The authors added some explanations, but I was also expected to find data comparisons by a statistical approach in the revised paper. Instead, they are missing in methods and results, and only descriptive comparisons have been provided.
Possibly, a different presentation of your research (e.g. by adding in the title "an explorative study", "descriptive study", or something of similar meanings ) might partially solve.
I also suggest further improving the presentation of the results: making all figures and tables "stand-alone"  by adding in a figure legend all the necessary information for reading and understanding the figures, without the need to search in the text.
Consistency between the text and the figure has to be improved (e.g., if you show Groups 1 to 3 in figure 6, where have they been previously presented in the text? In the text, you defined a group by robot and a group without. No mentions about group 1, 2 and 3)
Finally, a discussion of the results before the conclusions is still missing.

Author Response

Good afternoon,
Thank you for the comments made, we worked on the changes and here we attach the new document.

Regarding the changes, we believe that more statistical analysis cannot be made regarding the amount of people that were evaluated and considering that all four studies were applied considering completely different contexts. But a descriptive analysis is very likely to what we tried to achieve, so that is included in the Title of the new document.

Your comment of figure 6 was considered and now it specifies everything in boxes. We hope it helps to understand its meaning by itself now.

Finally, a discussion section was added before the conclusion. We do believe that with this new section we can clarify some of the questions that could arise from our study. It specifies how the four studies can be compared to make the conclusions.

All new changes were made using the track changes as suggested, so we hope this helps to identify the new changes made during this new stage.

Best regards,

Christian Fernando López Orozco 

Reviewer 2 Report

I appreciate you considering my privacy concerns. FYI, acknowledging the schools by name in the acknowledgements still exposes the participants to risk of identification. If the participants are aware the location is identified by name, that's ok. If they're not, then you should remove the names and reword.

Task makes a lot more sense to me than diligence. Thank you.

The manuscript still reads like it needs a stronger connection back to the existing literature at the end to show how this study adds to the research literature. 

Author Response

Good afternoon,
Thank you for the comments made, we worked on the changes and here we attach the new document.

Regarding the changes, we decided to remove the name of the schools also in the acknowledgements. As mentioned before, we do have permissions from the schools in general but we would like to avoid any recognition that they would have towards the students.

An additional section of discussion was added before the conclusions, hoping this could help understand the connection of the study as a whole.

Also, all new changes were made using the track changes as suggested, so we hope this helps to identify the new changes made during this new stage.

Best regards,

Christian Fernando López Orozco 
